# Direct screening for chromatin status on DNA barcodes in yeast delineates the regulome of H3K79 methylation by Dot1

**Hanneke Vlaming[1], Thom M Molenaar[1], Tibor van Welsem[1], Deepani W Poramba-Liyanage[1], Desiree E Smith[2], Arno Velds[3], Liesbeth Hoekman[4], Tessy Korthout[1], Sjoerd Hendriks[1], AF Maarten Altelaar[4,5], Fred van Leeuwen[1]***

[1]Division of Gene Regulation, Netherlands Cancer Institute, Amsterdam, Netherlands; [2]Department of Clinical Chemistry, Metabolic Laboratory, VU University Medical Center, Amsterdam, Netherlands; [3]Central Genomics Facility, Netherlands Cancer Institute, Amsterdam, Netherlands; [4]Mass Spectrometry/ Proteomics Facility, Netherlands Cancer Institute, Amsterdam, Netherlands; [5]Biomolecular Mass Spectrometry and Proteomics, Utrecht Institute for Pharmaceutical Sciences, University of Utrecht, Utrecht, Netherlands

*For correspondence: fred.v. leeuwen@nki.nl

**Abstract** Given the frequent misregulation of chromatin in cancer, it is important to understand the cellular mechanisms that regulate chromatin structure. However, systematic screening for epigenetic regulators is challenging and often relies on laborious assays or indirect reporter read-outs. Here we describe a strategy, Epi-ID, to directly assess chromatin status in thousands of mutants. In Epi-ID, chromatin status on DNA barcodes is interrogated by chromatin immunoprecipitation followed by deep sequencing, allowing for quantitative comparison of many mutants in parallel. Screening of a barcoded yeast knock-out collection for regulators of histone H3K79 methylation by Dot1 identified all known regulators as well as novel players and processes. These include histone deposition, homologous recombination, and adenosine kinase, which influences the methionine cycle. Gcn5, the acetyltransferase within the SAGA complex, was found to regulate histone methylation and H2B ubiquitination. The concept of Epi-ID is widely applicable and can be readily applied to other chromatin features.

## Introduction

The genome is packaged by histone proteins that are decorated with a wide variety of modifications. This provides a versatile marking mechanism that integrates cellular signals and plays a key role in processes such as transcription and DNA repair. Regulating the proper composition and modifications of chromatin is critical, as exemplified by the many cancer-promoting chromatin alterations and the epigenetic drugs in clinical trials (*Brien et al., 2016*).

At a first level, chromatin modification is under the control of modifying and demodifying enzymes, many of which have been identified. However, it is becoming increasingly clear that there is a second level of control that can involve a plethora of mechanisms, including targeting of enzymes to their site of action, cofactor availability, histone modification cross-talk, and regulation by protein-protein interactions. Understanding the establishment of epigenetic states and designing strategies to perturb them in treatment of disease will require a thorough and comprehensive understanding of the network of signals that feeds into the histone modification systems.

**eLife digest** To fit into the nucleus of eukaryotic cells (which include plant, animal and yeast cells), DNA wraps around histone proteins to form a structure called chromatin. Histones can be modified by a variety of chemical tags, which affect how easily nearby DNA can be accessed by other molecules in the cell. These modifications therefore help to control the activity of the genes encoded in the DNA and other key processes such as DNA repair.

If histone modifications are not regulated correctly, diseases such as cancer may result. Enzymes generally perform the actual modification, but there is another layer of regulation that controls the activity of these enzymes that not much is known about.

The activity of an enzyme that performs a histone modification known as H3K79 methylation (which involves a methyl chemical group being added to a particular region of a particular histone protein) has been linked to some forms of leukemia. Collections of mutant yeast cells can be used to identify the factors that regulate histone modifications in both yeast and human cells. However, current methods that screen for these regulators are time consuming.

To make the search for histone modification regulators more efficient, Vlaming et al. developed a new screening procedure called Epi-ID that can measure the amount of a specific histone modification in thousands of budding yeast mutants at the same time. In Epi-ID, each mutant yeast cell has a unique DNA sequence, or "barcode". The mutant cells are mixed together and the barcodes that are modified by a particular histone modification – such as H3K79 methylation – are isolated and then counted using a DNA sequencing technique. A high barcode count of a certain mutant indicates that more of the histone modification occurs in that mutant.

Using Epi-ID to survey H3K79 methylation enabled Vlaming et al. to successfully identify all previously known H3K79 methylation regulators, as well several new ones. These new regulators included enzymes that deposit histones on DNA, that carry out DNA repair, and that modify or de-modify histone proteins. To move forward with the newly identified regulators, it will be important to analyze how they control H3K79 methylation in yeast cells and to determine whether the regulators also control H3K79 methylation in human cells.

Finally, Epi-ID can be used to identify regulators of other types of histone modifications. A better understanding of chromatin regulation – and H3K79 methylation regulation in particular – can increase our understanding of diseases in which chromatin is deregulated, and may yield new strategies for the treatment of such diseases.

A specific histone modification of great clinical importance that is still poorly understood is methylation of histone H3 on lysine 79 (H3K79) (*Nguyen and Zhang, 2011*; *Vlaming and van Leeuwen, 2016*). H3K79 is methylated by Dot1/DOT1L. Human DOT1L is important in a subset of leukemias that express an MLL fusion protein caused by rearrangements of the MLL gene (MLL-r) (*Wang et al., 2016*). DOT1L inhibitors have been developed and are currently in clinical trials for the treatment of MLL-r leukemia (*Stein and Tallman, 2015*). DOT1L has also been implicated in other cancers, and it has a role in normal development and cellular reprogramming (reviewed in *McLean et al. (2014)* and *Nguyen and Zhang (2011)*). Furthermore, Dot1 plays a role in meiotic checkpoint activation and the DNA damage response (*Nguyen and Zhang, 2011*). Interestingly, the exact mechanism of action of H3K79 methylation remains poorly understood (*Vlaming and van Leeuwen, 2016*). The discovery of the important roles of H3K79 methylation has spiked an interest in the regulation of this modification. Both H3K79 methylation and the Dot1(L) enzyme are conserved from budding yeast to humans. Although some regulators, such as MLL fusion partners AF9 and AF10, are specific to DOT1L, other regulators are conserved throughout evolution (*Vlaming and van Leeuwen, 2016*). The best-characterized regulatory pathway is a trans-histone cross-talk: ubiquitination of H2B by Bre1 in yeast or RNF20/40 in mammals promotes H3K79 methylation (*Weake and Workman, 2008*). Other H3K79 methylation regulators may yet be discovered and could be potential new drug targets in diseases in which DOT1L has been implicated.

To search for regulators of H3K79 methylation we took advantage of the yeast knock-out collection. Thus far, systematic analysis of chromatin regulatory mechanisms has been challenging; the

available techniques are either laborious or indirect (see discussion). Here we describe a screening strategy to identify chromatin regulators in a quantitative, systematic, and high-throughput manner using a direct read-out of the chromatin status in mutant libraries. In this method, which we call Epi-ID, chromatin status on DNA barcodes in the yeast genome is directly interrogated by chromatin immunoprecipitation (ChIP) on a pool of cells and read out by parallel deep sequencing. With the Epi-ID protocol, it is possible to screen thousands of strains for effects on a chromatin feature of interest.

We used Epi-ID to screen the yeast knock-out collection for factors that influence H3K79 methylation (H3K79me). This screen identified all known regulators of H3K79me, as well as several new regulators that could be validated. We found that histone deposition and homologous recombination negatively regulate H3K79me and that adenosine kinase promotes H3 methylation through its effect on the methionine cycle. Finally, the histone acetyltransferase module in the SAGA complex was identified as an H3K79me regulator and subsequently shown to regulate H2B ubiquitination and other downstream methylation marks, probably through the stability of the deubiquitinating enzyme Ubp8, which is also a member of the SAGA complex.

In summary, Epi-ID is a direct, efficient and widely applicable screening technology. The Epi-ID screen presented here yielded a comprehensive picture of the H3K79 methylation regulome, and the technique can be readily applied to other chromatin modifications or chromatin-binding proteins.

## Results

### Epi-ID outline

DNA barcodes, unique sequences that can serve as identifiers, enable experiments on pools of cells (e.g. *Yan et al., 2008*). When counted by high-throughput sequencing, they yield quantitative information. Classically, collections of barcoded yeast knock-out strains have been used for competitive growth assays, e.g. to find genes that mediate drug toxicity or resistance (*Giaever and Nislow, 2014*). Here we take advantage of a yeast collection with barcodes in the genomic DNA, and thus packaged into chromatin, and use the barcodes to report on chromatin modification status or binding events. By performing ChIP on pools of barcoded cells and counting the abundance of the barcodes by high-throughput sequencing, the relative enrichment of each barcode can be determined. Since each barcode corresponds to a gene deletion, the enrichment of the barcode reports on the effect of the gene deletion on the abundance of the chromatin feature assessed by ChIP (*Figure 1A*). We have successfully tested this concept previously in combination with the recombination-induced tag exchange (RITE) assay to find regulators of histone turnover in a small set of candidates (*Verzijlbergen et al., 2011*). Now, we modified the technique such that it is applicable to screening the complete set of non-essential knock outs and applied Epi-ID to find regulators of H3K79 methylation by Dot1.

First, we generated a new barcoded yeast knock-out library by crossing a knock-out library that does not contain barcodes (*Tong and Boone, 2006*), with a Barcoder library (*Yan et al., 2008*) (*Figure 1A*). Barcoder strains (~1100) harbor two unique 20-base-pair barcodes at a common location. The barcodes are upstream (UpTag) and downstream (DownTag) of a selection marker (KanMX) integrated at a safe-harbor locus, the well-studied HO gene. We obtained a set of approximately 4300 strains, divided over five subsets with unique barcodes. With this new library, chromatin structure or modification status can be measured at a common locus in many knock-out strains, avoiding possible position effects of the integrated barcoded cassette. However, the UpTag and DownTag differ in their genomic contexts, being surrounded by promoters or terminators, respectively (*Figure 1A*). Thus, the two barcode positions will give information on Dot1 regulation in different functional contexts.

Second, we optimized the barcode-seq library construction (*Figure 1—figure supplement 1A*). The sequencing library was generated in a single round of amplification. UpTag and DownTag were amplified separately, using primers that annealed to common sequences immediately flanking the barcodes. An index introduced in this PCR allowed for extensive multiplexing, of up to at least 150 samples in one Illumina HiSeq lane.

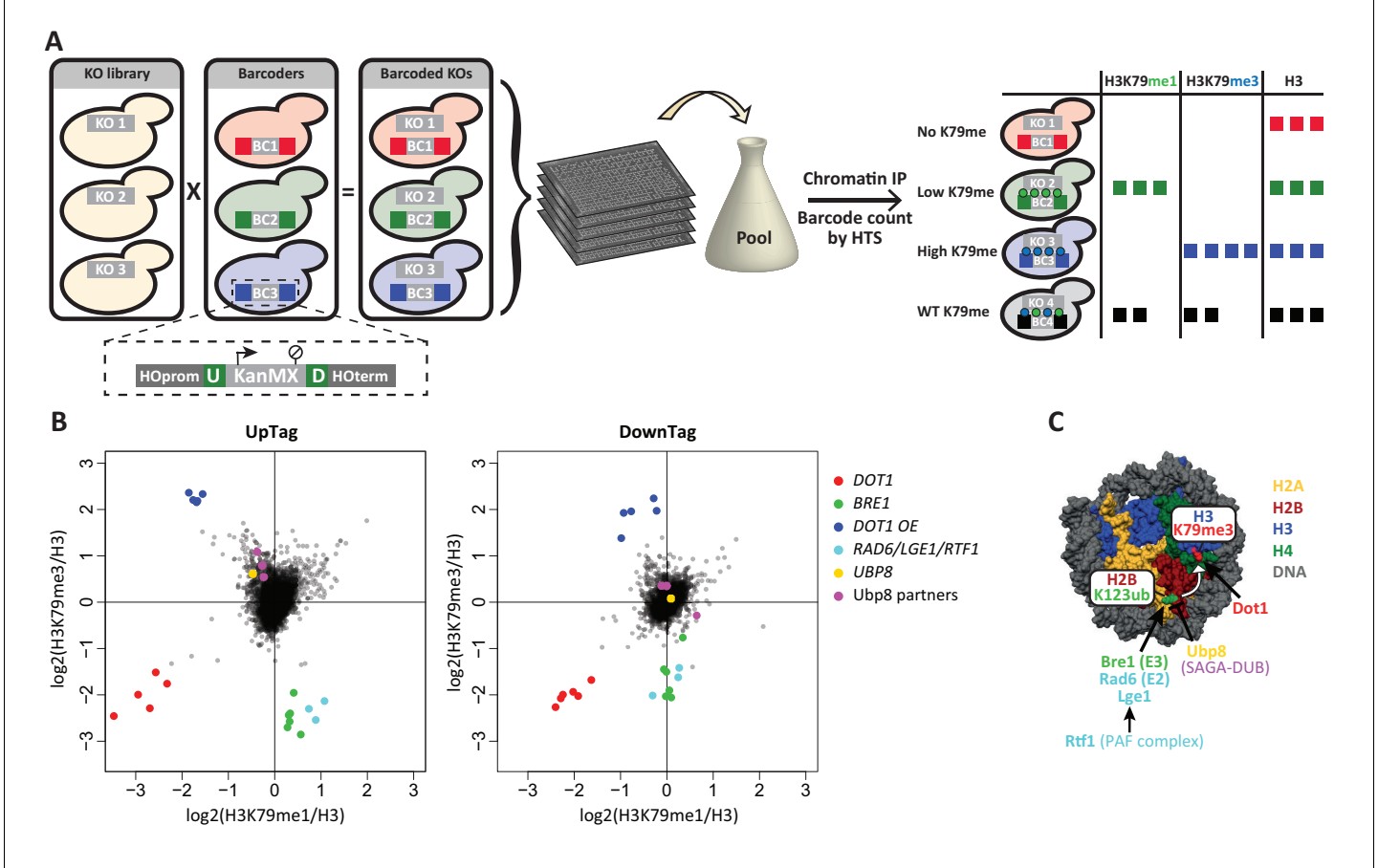

**Figure 1.** Outline and proof-of-concept of Epi-ID. (**A**) The barcoded knock-out library used for Epi-ID was created by crossing the NatMX knock-out library (**Tong and Boone, 2006**) with the Barcoder collection (**Yan et al., 2008**). Two 20-base-pair barcodes (UpTag (U) and DownTag (D)) flank a KanMX selection marker replacing the HO gene. For an Epi-ID experiment, barcoded mutant strains are pooled and ChIP experiments are performed on the pool. The barcodes of the different mutant strains can be counted by high-throughput sequencing and serve as a read-out for the amount of epitope present at the barcode. (**B**) Average data of two Epi-ID screens on approximately 4100 yeast deletion strains. Each dot represents a deletion strain, with the exception of the Dot1 overexpression strain (Dot1 OE). Control strains and some known regulators are highlighted. (**C**) Schematic depiction of the H2Bub pathway regulating H3K79 methylation.

The following source data and figure supplement are available for figure 1:

**Source data 1.** Epi-ID data.
**Figure supplement 1.** Technical details on Epi-ID.

Before performing the H3K79me Epi-ID, we confirmed the presence of H3K79 methylation around the barcodes by ChIP-qPCR in a wild-type strain. Both barcode loci showed an intermediate level of H3K79 methylation in a wild-type strain (*Figure 1—figure supplement 1B*). Here it is important to consider Dot1's distributive mechanism of methylation (*Frederiks et al., 2008*). This distributive mode of action leads to a characteristic shift in methylation states with changing Dot1 activity (*Figure 1—figure supplement 1C*). With increasing Dot1 activity, H3K79me1 will first increase and then decrease, as it is being converted into higher methylation states. Around the barcodes, H3K79me1 was high and H3K79me3 was low compared to coding sequences; this intermediate level of H3K79 methylation was consistent with the intergenic location of the barcodes and indicated that they should be able to report increased as well as decreased Dot1 activity (*Figure 1—figure supplement 1C*).

## Epi-ID finds the known H3K79 methylation regulators

We applied the optimized Epi-ID protocol to screen for regulators of H3K79 methylation. H3K79me1 and H3K79me3 were used for measuring changes in Dot1 activity since the combination of these methylation states provides the most informative and robust readout of Dot1 activity. Input DNA and total H3 ChIP were used for normalization purposes. All barcode count data can be found in *Figure 1—source data 1*.

The average data of two Epi-ID experiments was plotted as H3K79me1 versus H3K79me3, each normalized to H3 (*Figure 1B*). Data was median-normalized within each library subset, based on the assumption that most mutant strains have a wild-type level of H3K79 methylation. Uniquely-barcoded *dot1Δ*, *bre1Δ* and Dot1 over-expression control strains were added to each library subset as internal controls. The E3 ligase Bre1 ubiquitinates histone H2B on lysine 123, thereby promoting Dot1 activity, and in a *bre1Δ* strain H3K79 methylation is reduced (*Weake and Workman, 2008*). A Dot1 over-expression strain has high levels of methylation. The spiked-in controls were clear outliers: *dot1Δ* strains showed low H3K79me1 and H3K79me3 at both the UpTag and DownTag, *bre1Δ* strains showed low H3K79me3 and high H3K79me1, and Dot1 over-expression strains showed high H3K79me3 and low H3K79me1. The independent *dot1Δ* and *bre1Δ* strains present in the original library behave the same as their added counterparts. The results of the spiked-in control strains confirmed that Epi-ID can be used to identify strains with lower and higher levels of H3K79 methylation in pools of mutants.

Several other strong outliers could readily be explained, since they were known to affect H2B ubiquitination and H3K79 methylation (*Figure 1C*). Positive regulators of H3K79 methylation were Rad6 and Lge1, which form the H2B ubiquitination complex together with Bre1 (*Weake and Workman, 2008*), and Rtf1, which is part of the PAF transcription-elongation complex and recruits Bre1/Rad6 to chromatin of transcribed regions (*Piro et al., 2012*). Ubp8 and its partners in the deubiquitinase (DUB) module of the SAGA complex (Sgf73, Sgf11 and Sus1) together deubiquitinate H2B and predominantly act at the 5′ ends of transcribed regions (*Bonnet et al., 2014*; *Morgan et al., 2016*; *Schulze et al., 2011*). In the Epi-ID screen, deletion of the genes encoding these proteins led to increased methylation on the UpTag, but not on the DownTag, as expected given their respective promoter and terminator context. Notably, deletion of the other H2B DUB, *UBP10*, did not increase H3K79me on the barcodes, consistent with the observation that Ubp10 preferentially acts on telomeres (*Gardner et al., 2005*). In summary, Epi-ID identified all established H3K79 methylation regulators acting via H2Bub in euchromatin.

## Deposition of new histones counteracts H3K79 methylation in the absence of a demethylase

H3K79 methylation is very stable and accumulates on old histones (*De Vos et al., 2011*). No H3K79 demethylases are known (*Sweet et al., 2010*; *Zee et al., 2010*) and the Epi-ID screen provided no evidence of an H3K79 demethylase either. None of the genes with a known demethylase domain was found among the negative regulators (*Figure 2—figure supplement 2*). Given the likely absence of a demethylase, the main mechanism to counteract Dot1 activity may be dilution of methylated histones. This can occur by replication-independent turnover of histones (*Zentner and Henikoff, 2013*), or by dilution during S phase when the biggest influx of new histones occurs. A negative correlation between histone turnover and H3K79me3 level has been observed in genome-wide maps (*Radman-Livaja et al., 2011*; *Weiner et al., 2015*). Based on these observations and computational modeling, it has been proposed that slow-growing cells accumulate more H3K79 methylation (*De Vos et al., 2011*). To test this hypothesis without having to change temperature or carbon source, we used barcode-seq to derive growth rates for each of the knock-out clones in the pool of cells (*Figure 2—source data 1* and *Figure 2—figure supplement 1*) and determined the relation between growth rate and H3K79me level. As a single score for H3K79 methylation level, we used the H3K79me3/H3K79me1 ratio. This score was very robust, with a Pearson correlation of 0.89 between two screens. *Figure 2A* shows a negative correlation between the growth rate and the H3K79 methylation score on both UpTag and DownTag (r = −0.4) and methylation was significantly higher for the slow growers than for the remainder of the yeast strains (*Figure 2A*). It can also be appreciated from this plot that the known regulators identified in *Figure 1* remain outliers after taking into account the growth defects that some of them have.

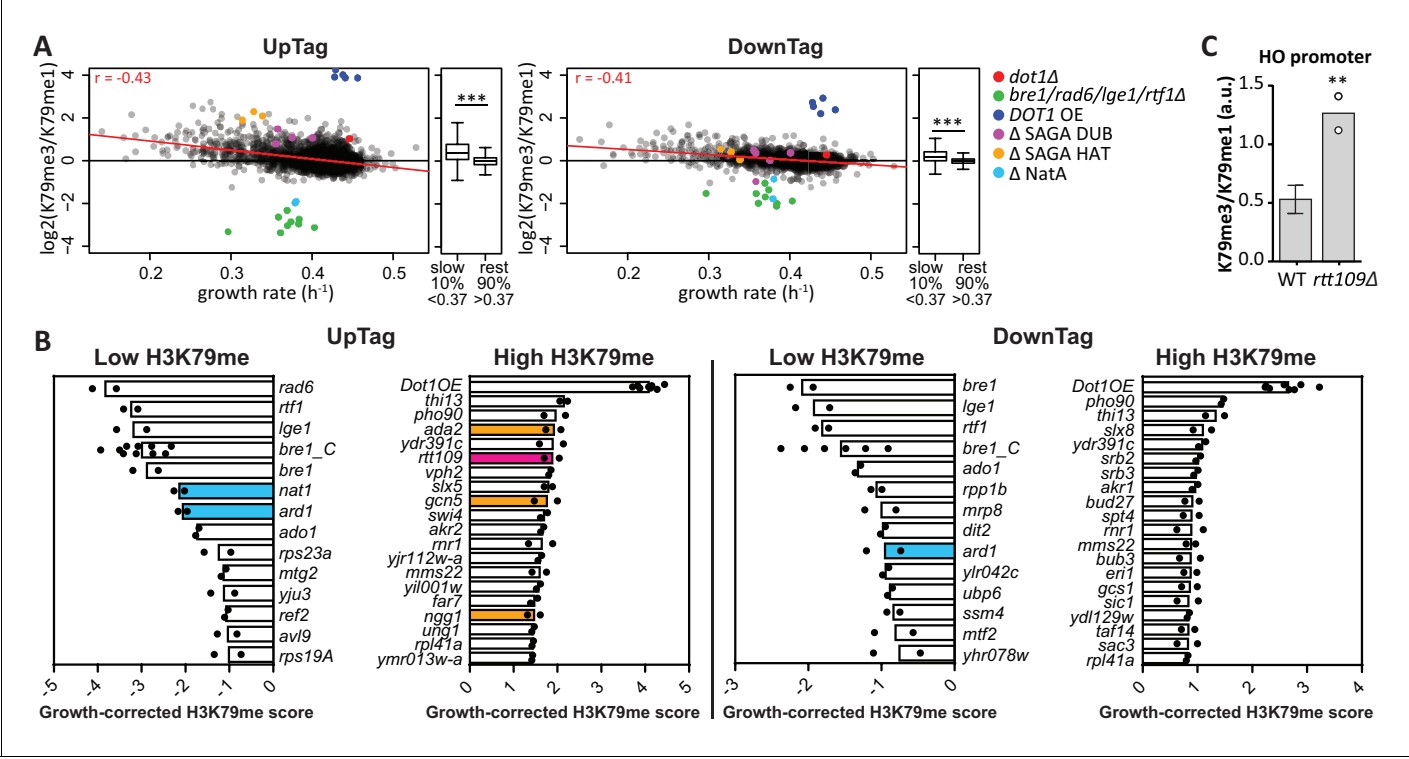

**Figure 2.** H3K79 methylation regulation by growth and acetyltransferases. (A) Scatter plots of growth rate and H3K79 methylation (me3/me1) for UpTag and DownTag, each dot representing a mutant strain. The Pearson correlation coefficient is shown in the plot. The red line is the linear model fitting the data best and was used for correcting the H3K79 methylation score. Highlighted strains were ignored in the analysis, because they lacked validated H3K79me regulators. Alongside the scatter plot is a Tukey box plot to compare the median and spread of H3K79 methylation for the bottom-10% slowest growers, compared to the other 90%. These populations are highly significantly different, as determined by a T test. (B) Bar charts of growth-corrected H3K79 methylation scores of deletion strains, showing the strongest positive and negative outliers on UpTag and DownTag. Because individual outliers are shown, a cutoff was applied on variation between the two biological replicates (c.o.v. <0.35) to increase confidence. Mean and individual data points of two experiments are shown. Strain *bre1_C* is the control *bre1Δ* strain that was taken along multiple times in each experiment, whereas the other *bre1Δ* strain was part of the library. Strain *rtt109Δ* and NatA complex mutants have been highlighted, as well as mutants of the SAGA HAT module that will be discussed later on. (C) ChIP-qPCR analysis at the HO promoter, near the UpTag. Plotted is the ratio between H3K79me3 and H3K9me1 IP values. Four wild-type (mean with SD) and two *rtt109Δ* strains (individual data points shown) were compared by an unpaired T test.

The following source data and figure supplements are available for figure 2:

**Source data 1.** Growth rates calculated for all deletion strains.

**Source data 2.** Growth-corrected H3K79me scores.

**Source data 3.** ChIP-qPCR data at the HO promoter, WT vs *rtt109Δ*.

**Figure supplement 1.** Growth rate determination.

**Figure supplement 2.** Growth-corrected H3K79me scores of genes that contain potential demethylase domains.

Having determined the relationship between growth and H3K79 methylation, we calculated a growth-corrected methylation score for each strain, by dividing the H3K79me3/H3K79me1 ratio by the methylation shift expected for the fitness of the strain (*Figure 2—source data 2*). The expected shift was based on the linear model fitting the data (the red line in *Figure 2A*). The robust top outliers based on this growth-corrected methylation score are shown in *Figure 2B*. Rtt109 was one of the strongest negative regulators of H3K79 methylation on the UpTag (*Figure 2B*), and high H3K79 methylation upon deletion of *RTT109* could be validated by ChIP-qPCR (*Figure 2C*). Rtt109 is a histone acetyltransferase that acetylates newly synthesized histone H3 on lysine 56 (*Driscoll et al.,*

2007; *Han et al., 2007*). Through this activity, Rtt109 promotes histone transport and nucleosome assembly (*Dahlin et al., 2015*). *RTT109* deletion directly leads to decreased turnover at 'hot' nucleosomes, mostly found in promoters (*Dion et al., 2007*; *Kaplan et al., 2008*). The fact that Rtt109 was one of the strongest negative regulators of H3K79me at the UpTag, i.e. in a promoter region, indicates that histone turnover is an important determinant of the H3K79me level. Altogether, these data support the idea that no H3K79 demethylase is active in yeast and show that the deposition of new histones (replication-coupled or -independent) is an important mechanism to counteract H3K79 methylation.

## The NatA Complex regulates H3K79 methylation and H2B ubiquitination

Among the strongest positive regulators of H3K79me on both the UpTag and DownTag were Nat1 and Ard1, the two components of the NatA N-acetyltransferase complex. The DownTag score of the *nat1Δ* strain was filtered out in *Figure 2B* based on its variation between replicates, but it was a positive regulator as well. Ard1 has been reported to promote H2Bub and specifically H3K79me3, but the role of Nat1 remained uncertain (*Takahashi et al., 2011*). We confirmed the effect of Ard1 on H2B ubiquitination and H3K79 methylation, and found an identical effect for Nat1 (*Figure 3A*). Also H3K4me3 and H3K36me3 were decreased in *nat1Δ* and *ard1Δ* strains, and again the effect was partial compared to the *bre1Δ* strain (*Figure 3A*). H3K4me3 is known to depend on H2B ubiquitination (*Dover et al., 2002*), but the decrease in H3K36me3 we observed in the *bre1Δ* strain was not reported before. We confirmed the decrease in H3K36me3 in the absence of H2B ubiquitination (*Figure 3—figure supplement 1C*) and observed that H3K36me2 was not affected. We conclude that the NatA complex is required for a normal H2Bub level and thereby promotes all downstream methylation events. Notably, that NatA acts upstream of Dot1 is in agreement with our previous observation that NatA and Dot1 act in the same silencing pathway (*van Welsem et al., 2008*).

NatA acetylates about one third of the proteins encoded in the yeast genome and may affect their stability or interactions to other proteins (*Aksnes et al., 2016*). We checked whether altered protein levels of Dot1 and H2Bub regulators, all of which are putative NatA targets based on sequence (*Aksnes et al., 2016*), could explain the reduced H3K79me and H2Bub abundance in *nat1Δ* strains. However, no significant decrease in Dot1 and H2B ubiquitination factors or increase in H2Bub deubiquitinating factors was observed (*Figure 3—figure supplement 1D–G*). Fully understanding which of the many N-acetylated proteins are responsible for the role of NatA in control of H2B ubiquitination and the downstream methylation events will require a comprehensive mutational analysis of the yeast N-terminal proteome.

## Adenosine kinase promotes histone methylation

Another strong positive H3K79me regulator was *ADO1*. At both the UpTag and DownTag the *ado1Δ* strain showed very low H3K79 methylation. Immunoblot analysis confirmed *ADO1* as a positive regulator of H3K79 methylation, showing that also on a global level there is a shift from H3K79me3 to H3K79me1 in the *ado1Δ* strain (*Figure 3B*). In addition, H3K4me3 and H3K36me3 were decreased in *ado1Δ* cells (*Figure 3B*). To rule out that the methylation decreases were caused by low H2B ubiquitination, H2Bub levels were determined by immunoblot. H2Bub was not decreased in *ado1Δ* cells (*Figure 3B*), if anything it was somewhat increased, which could mask a stronger methylation defect (*Figure 3—figure supplement 2A,B*). Dot1 protein expression was not altered in the *ado1Δ* strain (*Figure 3—figure supplement 2C*).

The *ADO1* gene encodes yeast adenosine kinase, responsible for phosphorylating adenosine to generate AMP. Clearance of adenosine by Ado1 may indirectly affect the methylation cycle and thereby the methylation potential in the cell (*Boison, 2013*; *Figure 3C*). Disruption of the methionine cycle by genetic perturbation or nutrient supply is known to affect histone methylation in yeast and other organisms (*Janke et al., 2015*; *Sadhu et al., 2013*). To investigate the role of adenosine kinase in the SAM cycle, we determined SAM and SAH levels in *ado1Δ* and wild-type cells. As shown in *Figure 3D*, SAH levels increased 19-fold in the absence of adenosine kinase. This was expected based on the reaction scheme shown in *Figure 3C*, and is in agreement with the lower histone H3 methylation levels (*Figure 3B*), since SAH is a known inhibitor of methyltransferases (*Etchegaray and Mostoslavsky, 2016*; *Richon et al., 2011*). However, SAM levels increased by as

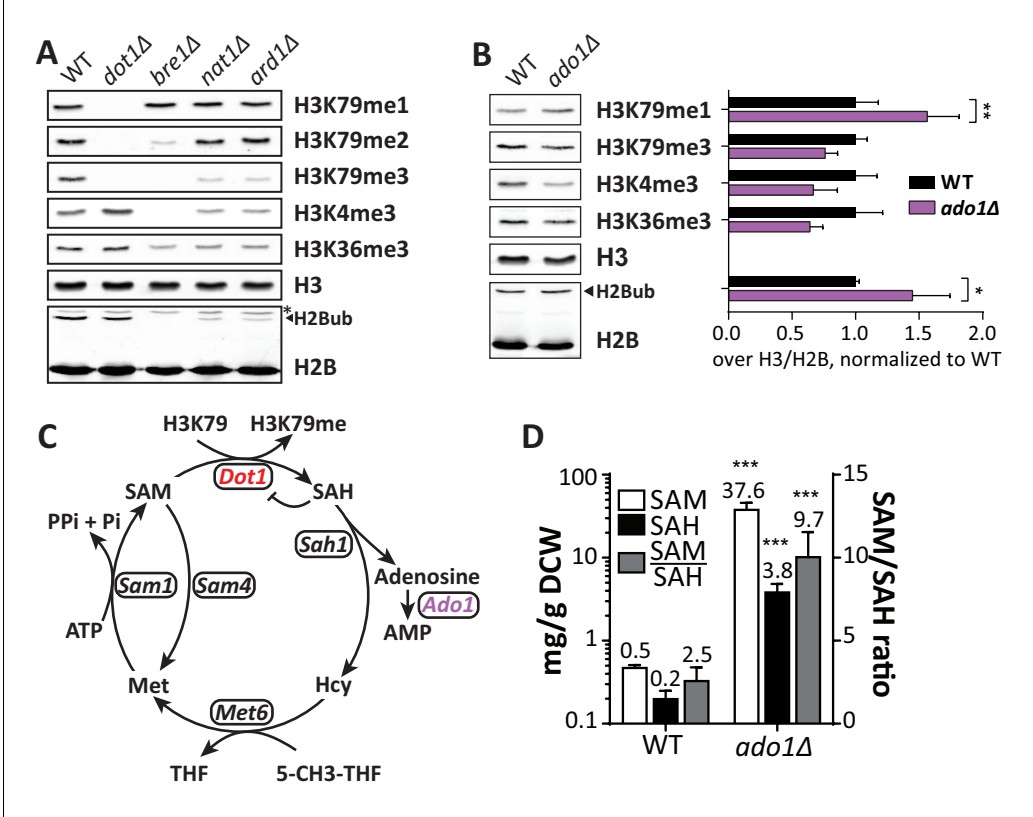

**Figure 3.** Positive regulators of H3 methylation. (**A**) Immunoblots of H3 methylation and H2B in the indicated strains. Biological replicates are shown in *Figure 3—figure supplement 1A*. (**B**) Representative image of immunoblots of H3 methylation and H2B in wild-type and *ado1Δ* strains. Alongside is the quantification of the three biological replicates run on these gels. Uncropped blots are shown in *Figure 3—figure supplement 2A*, more replicates in 2B. (**C**) Schematic depiction of the methionine cycle in budding yeast. Dot1 and Ado1 have been highlighted. In addition to Dot1, SAM is utilized by many other methyltransferases. (**D**) Levels of SAM and SAH on a logarithmic scale, and their ratio on a linear scale. n = 7–8, comparison by multiple T test.

The following source data and figure supplements are available for figure 3:

**Source data 1.** Raw SAM and SAH measurements and calculations.
**Figure supplement 1.** NatA immunoblots.
**Figure supplement 2.** Ado1 immunoblots.

much as 75-fold, causing an unexpected 4-fold increase of the SAM/SAH ratio. The increase in SAM levels may be caused by a global inhibition of methyltransferases by excess SAH and therefore reduced SAM usage, or by compensatory mechanisms such as altered expression of methionine bio-synthesis genes (*Kanai et al., 2013*). Although the SAM/SAH ratio is often considered to determine the activity of methyltransferases (*Etchegaray and Mostoslavsky, 2016*), our findings suggest that the absolute levels of SAH and SAM are also important.

## DNA repair complexes regulating H3K79 methylation

To find the more subtle second-level regulators of H3K79me in a systematic and unbiased manner, we first used Cutoff Linked to Interaction Knowledge (CLIK) to determine the threshold above which outliers were likely to be genuine regulators (*Dittmar et al., 2013*). CLIK is based on the notion that the rate of interactions among true positives in a screen is higher than for a random set of genes. The algorithm indeed identified many genetic and/or physical interactions between genes at both

the low methylation and high methylation ends of the data, for both UpTag and DownTag (*Figure 4—figure supplement 1*), and determined cutoffs based on the points in the rank list where interaction density dropped. The groups defined in this way were considered candidate regulators.

Rather than focusing on individual genes in these groups, we took two bioinformatics approaches to identify complexes and processes that regulate H3K79 methylation. First, the built-in complex enrichment analysis in the CLIK tool identified enriched complexes in all groups of candidate regulators (*Table 1* and *Table 1—source data 1*). Few new complexes were identified among the positive regulators, so we focused our attention on the negative regulators. Strikingly, of the enriched complexes on the UpTag, Mms22-Rtt101-Mms1, Slx5-Slx8, and Rad51-Rad57 were all involved in DNA repair, specifically at replication forks (see below) (*Branzei et al., 2006*; *Costes and Lambert, 2012*; *Duro et al., 2008*; *Su et al., 2015*). Second, we used PANTHER (*Mi et al., 2016*) to identify processes enriched among the candidate negative regulators at the UpTag (*Table 2*). With the exception of two very general terms, all enriched GO processes fall into the DNA repair category. The highest enrichment scores were found for double-strand break (DSB) repair via homologous recombination (HR) and its parent recombinational repair (*Table 2*, *Figure 4A*).

**Table 1.** Enriched complexes in the candidate regulator groups (thresholds determined by the CLIK tool), on UpTag and DownTag. Complex enrichment was determined by the built-in complex enrichment tool on the CLIK website. Within each group, enriched complexes are ranked by p value, only complexes with a p value below 0.01 are shown. For complexes with an asterisk, all components present in the data were found in the CLIK group.

| Complex | Candidate regulators | Not in candidate regulator group | Not in dataset | P value |
|---|---|---|---|---|
| **Low methylation (positive regulators) on UpTag (top-49 of 4231)** | | | | |
| Lge1/Bre1 complex * | 2 | 0 | 0 | 1.2E-04 |
| **Low methylation (positive regulators) on DownTag (top-64 of 4238)** | | | | |
| Lge1/Bre1 complex * | 2 | 0 | 0 | 2.1E-04 |
| Proteasome complex | 2 | 8 | 3 | 0.009 |
| $H^+$-transporting ATPase | 2 | 8 | 3 | 0.009 |
| **High methylation (negative regulators) on UpTag (top-247 of 4231)** | | | | |
| Rad51-Rad57 * | 5 | 0 | 0 | 4.9E-07 |
| SAGA complex | 4 | 5 | 13 | 9.2E-04 |
| Slx5/Slx8 complex * | 2 | 0 | 0 | 0.003 |
| Mms22/Rtt101/Mms1 complex * | 2 | 0 | 1 | 0.003 |
| Mdm12/Mmm1/Mdm10 complex * | 2 | 0 | 1 | 0.003 |
| ER V-ATPase assembly complex * | 2 | 0 | 0 | 0.003 |
| DNA-directed RNA polymerase II, holoenzyme | 3 | 4 | 8 | 0.005 |
| SWI/SNF complex | 3 | 4 | 5 | 0.005 |
| **High methylation (negative regulators) on DownTag (top-274 of 4238)** | | | | |
| Mediator complex | 6 | 4 | 14 | 1.3E-05 |
| Kornberg's mediator (SRB) complex | 6 | 5 | 14 | 2.7E-05 |
| Rpd3L complex | 5 | 4 | 3 | 1.2E-04 |
| Slx5/Slx8 complex * | 2 | 0 | 0 | 0.004 |
| Mms22/Rtt101/Mms1 complex * | 2 | 0 | 1 | 0.004 |
| Actin cytoskeleton-regulatory complex * | 2 | 0 | 1 | 0.004 |
| Bub1/Bub3 complex * | 2 | 0 | 0 | 0.004 |
| Kinetochore | 3 | 4 | 11 | 0.008 |

Source data 1. CLIK groups and complexes enriched in each group.

**Table 2.** Enriched processes in the group of candidate negative regulators on the UpTag. Enrichment determined by PANTHER (**Mi et al., 2016**). Terms are organized by hierarchy; daughter terms are indented. P values were Bonferroni-corrected.

| GO biological process complete | Enrichment | p value |
|---|---|---|
| DNA metabolic process (GO:0006259) | 2.3 | 5.2E-03 |
| cellular response to DNA damage stimulus (GO:0006974) | 2.9 | 1.8E-04 |
| > DNA repair (GO:0006281) | 3.2 | 4.6E-05 |
| > > double-strand break repair (GO:0006302) | 4.4 | 8.8E-04 |
| > > recombinational repair (GO:0000725) | 5.9 | 1.2E-04 |
| > > > double-strand break repair via homologous recombination (GO:0000724) | 5.5 | 5.2E-03 |
| chromosome organization (GO:0051276) | 2.3 | 6.3E-04 |
| macromolecular complex subunit organization (GO:0043933) | 1.8 | 7.7E-03 |

To investigate this further, we performed validation experiments for the strongest individual outliers. Slx5 and Slx8 form a heterodimeric SUMO-targeted Ubiquitin Ligase complex (StUbL) that is required to maintain genomic stability (**Nagai et al., 2008**; **Xie et al., 2007**). Strains lacking Slx5 or Slx8 grow very slowly due to the amplification of endogenous 2μ circles (**Burgess et al., 2007**), so we first relieved the growth defect of these strains by curing them (and a wild-type control) of the 2μ plasmid (**Figure 4—figure supplement 2A**). Because the substrates of the Slx5/Slx8 StUbL complex largely overlap with those of the SUMO ligase Mms21 (**Albuquerque et al., 2013**), we also generated a strain in which the SUMO ligase activity of Mms21 was compromised (**Zhao and Blobel,**

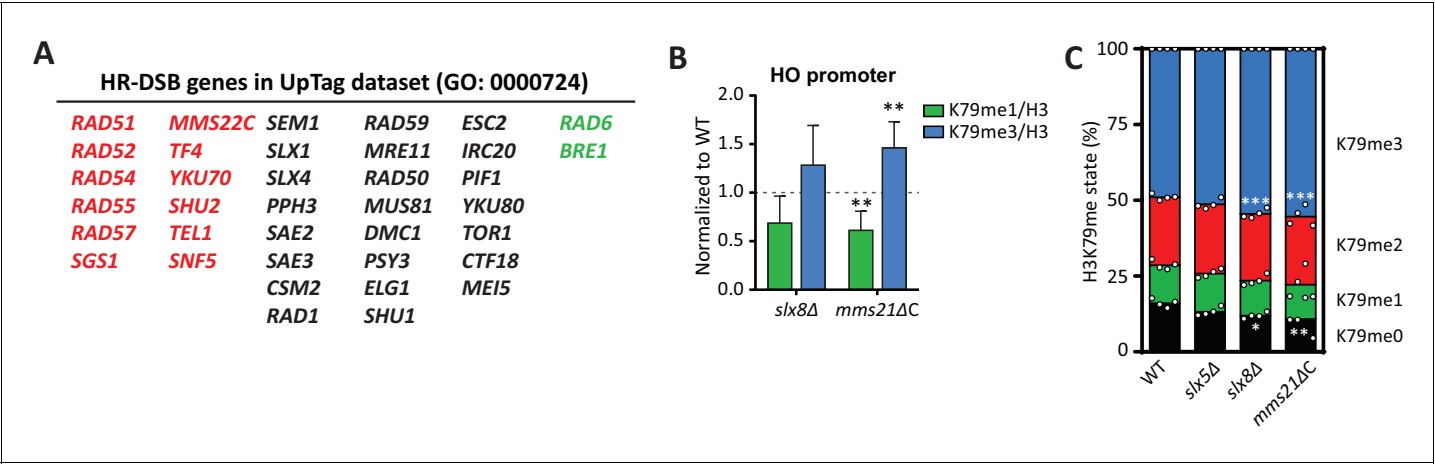

**Figure 4.** DNA repair genes are negative regulators of H3K79 methylation. (**A**) Genes from the GO process of double-strand break repair by homologous recombination, which are present in the dataset. Genes from the CLIK group of negative regulators are on the left (in red), positive regulators on the right (in green). (**B**) ChIP-qPCR analysis of H3K79 methylation on the HO promoter, close to the UpTag. K79me IPs were normalized to H3; mutants were normalized to wild-type. 4–6 replicates per strain, compared to one by one-sample T test. (**C**) MS analysis of H3K79 methylation levels in wild type and mutants. Mean and individual data points are shown.

The following source data and figure supplements are available for figure 4:

**Source data 1.** Normalized ChIP-qPCR data on HO promoter.
**Source data 2.** Raw mass spectrometry data.
**Figure supplement 1.** CLIK plots.
**Figure supplement 2.** Excluding mechanisms of H3K79me regulation by Slx5, Slx8 and Mms21.

*2005*). As shown in *Figure 4B*, ChIP-qPCR showed an increase in methylation at the HO promoter, near the UpTag, in both *slx8Δ* and *mms21ΔC*, implicating the SUMO-ubiquitin pathway in regulating H3K79 methylation. A small global increase in methylation was observed in these strains by mass spectrometry (*Figure 4C*), suggesting that the effect was not limited to the barcode regions.

No DNA-damaging agents were present in the Epi-ID experiment. However, also in untreated cells homologous recombination is needed to restart replication forks that have stalled or collapsed, for instance at structure-forming sequences or replication-fork barriers (*Costes and Lambert, 2012*). Mms22, Rtt101 and Mms1 (of which Rtt101 was not in the dataset) form a Cul4-like E3 ubiquitin ligase that sits at replication forks and promotes restart of stalled forks, presumably through homologous recombination (*Buser et al., 2016*; *Vaisica et al., 2011*). Rad52 and its partners are central players in homologous recombination, also at replication forks (*Costes and Lambert, 2012*). The Slx5 and Slx8 StUbL complex and Mms21 promote translocation of DNA breaks and collapsed replication forks to nuclear pores (*Burgess et al., 2007*; *Horigome et al., 2016*; *Su et al., 2015*). Interestingly, also the nuclear pore component required for this translocation, Nup84 (*Su et al., 2015*), was identified as a negative regulator of H3K79me in the Epi-ID screen. At the pore, Slx5/8 modulate homologous recombination by targeting protein substrates for degradation, including Rad52 itself (*Su et al., 2015*). Other factors from the process of HR-mediated DSB repair that were among the negative regulators (*Figure 4A*) include Sgs1, which also facilitates re-initiation of stalled replication forks (*Ashton and Hickson, 2010*), and Ctf4, which tethers Mms22 to the replisome (*Buser et al., 2016*). Taken together, among the second-level regulators of H3K79me are many factors required to allow recovery of stalled replication forks. This suggests that this process regulates H3K79me, either because the repair itself counteracts methylation or inhibits Dot1, or because unrepaired collapsed replication forks lead to accumulation of H3K79 methylation.

Since H2B ubiquitination has been described to facilitate fork recovery (*Lin et al., 2014*), we wondered if the regulation of H3K79 methylation could be through H2B ubiquitination. However, H2Bub levels were not increased (*Figure 4—figure supplement 2B*), and neither was Dot1 expression (*Figure 4—figure supplement 2C*). Unraveling the mechanism of this regulation will require further studies. Another interesting point is the function of this regulation. Dot1 has been shown to function in several DNA repair pathways; it mediates checkpoint activation and recombinational repair after UV damage, and represses translesion synthesis (*Conde and San-Segundo, 2008*; *Rossodivita et al., 2014*). Dot1 has not yet been studied in the context of replication fork stalling, but our finding that H3K79 methylation is regulated in this context warrants further investigations. Finally, since homologues of all involved proteins can be found in mammals, it will be interesting to see if also the described H3K79me regulation is conserved.

## The SAGA HAT module negatively regulates histone methylation and H2B ubiquitination

SAGA complex subunits were highly enriched among the negative H3K79me regulators on the UpTag (*Table 1*). These not only included subunits of the deubiquitinase (DUB) module, but also subunits of the histone acetyltransferase (HAT) module. Indeed, Gcn5 and its partners in the HAT module, Ada2 and Ngg1 (Ada3), were among the strongest negative regulators (*Figure 2B*). No data was obtained on Sgf29, the last SAGA HAT module subunit. The specificity these factors showed for the UpTag is not surprising given the activity of Gcn5 at promoters (*Bonnet et al., 2014*). Mass spectrometry measurements demonstrated that the effect of Gcn5 was not limited to the UpTag, since a *gcn5Δ* strain also had more H3K79me3 and less H3K79me1 globally (*Figure 5A*). To investigate the role of Gcn5 in more detail, we analyzed the behavior of a specific point mutant, Gcn5-F221A. This mutant protein has previously been shown to be catalytically inactive (*Kuo et al., 1998*; *Lanza et al., 2012*). To allow for a direct, quantitative, and sensitive comparison, we performed a custom-designed Epi-ID experiment with a set of *gcn5Δ* strains with single-copy plasmids. In contrast to wild-type Gcn5, the Gcn5-F221A mutant was not able to rescue the knock-out phenotype (*Figure 5B*; *Figure 5—figure supplement 1A,C*). This finding suggests that H3K79me regulation by Gcn5 depends on its catalytic activity. However, it cannot be excluded that the F221A mutation acts by other mechanisms such as disrupting nucleosome binding or altering the stability of Gcn5.

Dot1 protein expression was unaffected by deletion of *GCN5* (*Figure 5E*). Since Gcn5 is found in the SAGA complex, just like the H2Bub DUB Ubp8, we wondered if Gcn5 could function through

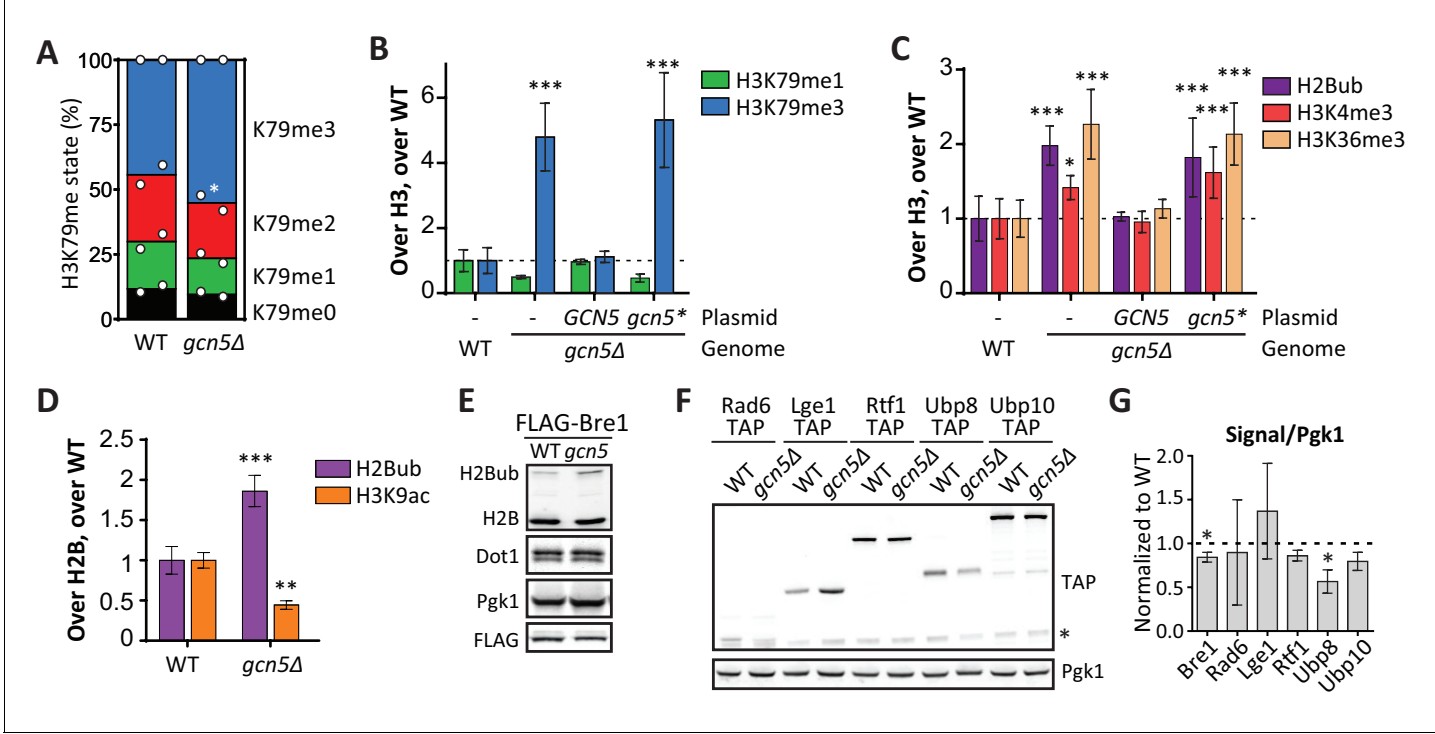

**Figure 5.** Gcn5 regulates H2B ubiquitination and H3 methylation. (**A**) MS analysis of H3K79 methylation levels in wild-type and *gcn5Δ* strains. Mean and individual data points of two biological replicates. (**B**) Custom Epi-ID experiment on strains harboring empty or *GCN5*-encoding CEN plasmids, grown in YC-LEU. Gcn5* contains the F221A mutation, abrogating catalytic activity. H3K79me1/H3 and H3K79me3/H3 are normalized to the mean of ten wild-types, five or six replicates per mutant. (**C**) Custom Epi-ID results for H2Bub, H3K4me3 and H3K36me3, from the same experiment as shown in panel **B**. (**D**) ChIP-qPCR analysis of H2Bub and H3K9ac, normalized to H2B, at the HO promoter near the UpTag. n = 3 each, the wild-type average was set to 1. (**E**) Immunoblots of a *GCN5+* and *gcn5Δ* strain, showing the levels of (ubiquitinated) H2B, Dot1 and Pgk1 as a loading control. The FLAG blot shows expression levels of Bre1, which was N-terminally FLAG-tagged in these strains. Replicate blots can be found in *Figure 5—figure supplement 2A*. (**F**) TAP blots were used to measure the expression levels of C-terminally TAP-tagged versions of the indicated proteins. Representative blot; note that the Rad6-TAP band is only just above the non-specific band indicated with an asterisk. (**G**) Quantification of the FLAG and TAP blots shown in panels **E** and **F**, as well as two TAP blots from independent experiments.

The following source data and figure supplements are available for figure 5:

**Source data 1.** Raw mass spectrometry data.

**Source data 2.** Data from custom Epi-ID experiment.

**Source data 3.** Data from H2Bub and H3K9ac ChIP-qPCR experiment.

**Figure supplement 1.** Custom Epi-ID experiments with controls.

**Figure supplement 2.** Immunoblots confirming the effect of Gcn5 on global H2Bub and Ubp8 levels.

**Figure supplement 3.** Gcn5 regulates H2Bub and H3K9ac at several loci.

H2B ubiquitination. Indeed, *gcn5Δ* strains were found to have significantly more H2Bub on the UpTag than wild-type strains, and this was also the case for cells expressing the inactive Gcn5-F221A protein (*Figure 5C*). The effect of Gcn5 on the H2Bub level at this locus was comparable to the effect of Ubp8 (*Figure 5—figure supplement 1B*). Since H2Bub levels were affected, we also tested the effect on H3K4me3 and H3K36me3 in the custom Epi-ID experiment and found both to be increased in the absence of Gcn5 acetyltransferase activity (*Figure 5C*). Thus, at the UpTag we

found the activity of Gcn5 to negatively regulate H2B ubiquitination and, probably through its ubiquitination effect, methylation of H3K4, H3K36 and H3K79.

To validate the H2Bub increase caused by *GCN5* deletion by an independent method and to test whether the effect was not limited to the barcode region, we took two approaches. First, using ChIP-qPCR, both an increase in H2Bub and a decrease in H3K9ac could be observed at the HO promoter, near the UpTag in *gcn5Δ* strains (*Figure 5D*). These changes were also observed at several other loci, but not at all of the loci examined (*Figure 5—figure supplement 3*). In the future, it will be interesting to determine where H2Bub regulation by Gcn5 occurs and how the specificity for certain regions is established. Second, immunoblotting was used to determine the global change in H2Bub level. Both *gcn5Δ* and *ada2Δ* strains showed increased H2Bub -levels (*Figure 5E*, *Figure 5— figure supplement A–C*).

Gcn5 may modulate H2Bub by acetylating histones or by interactions between the HAT and DUB modules. Indeed, acetylation on one of the DUB module components has been reported (*Henriksen et al., 2012*). Furthermore, interactions between the HAT module and the DUB module have also been observed by independent approaches in yeast and human (*Atanassov et al., 2009*; *Durand et al., 2014*; *Han et al., 2014*). To investigate this further, we examined the expression of H2Bub regulators and observed that Ubp8 protein expression was significantly decreased in the absence of Gcn5 (*Figure 5E–G*). A reduction in Ubp8 protein level was also observed in chromatin fractions (*Figure 5—figure supplement 2D–E*). Thus, the lower Ubp8 levels can at least in part explain the observed change in H2Bub in *gcn5Δ* cells. These findings suggest that Gcn5 is required for Ubp8 stability, potentially by regulating the association with its partners in the SAGA DUB module. Taken together, we identified the SAGA HAT as a negative regulator of H3K79 methylation, and then discovered that it regulates H2B ubiquitination and subsequent methylation events on three lysine residues on histone H3, at least in part by maintaining Ubp8 protein expression. The sensitivity of the Epi-ID technique allowed for identification of this regulatory mechanism, even though the global effects are small and previously went unnoticed (*Lee et al., 2005*). Our finding that the SAGA HAT module regulates the DUB, together with the finding that the DUB module regulates HAT activity (*Han et al., 2014*), suggests that the two activities in the SAGA complex are highly interactive, despite being organized in distinct modules.

## Discussion

Chromatin regulatory mechanisms are important for genome function and are often deregulated in cancer and other diseases (*Brien et al., 2016*). To enable efficient and direct screening for chromatin regulators, we developed Epi-ID, in which chromatin modification or binding-events at DNA barcodes are directly interrogated by ChIP followed by deep sequencing.

The Epi-ID technique has several advantages over the methods that are currently available. First, in contrast to other techniques used to screen for chromatin regulators (*Dover et al., 2002*; *Peng et al., 2008*; *Schulze et al., 2009*), Epi-ID can be performed on pools of cells, making it easily applicable to large collections of mutants. Second, Epi-ID is a direct method, in contrast to reporter gene assays that have been used to read out chromatin changes (*Rossmann et al., 2011*). Third, it is a very sensitive method that is able to pick up differences that cannot be detected by immunoblot. Finally, by comparing all knock-outs directly in a competitive manner, no complex normalization methods are required.

As a proof of concept we applied Epi-ID to delineate the regulome of H3K79 methylation in budding yeast. All known H3K79 methylation regulators were identified, as well as several new regulators. All tested candidate regulators could be validated on a global level and/or locally, by ChIP-qPCR. It is clear from our data that the well-studied H2Bub-H3K79me trans-histone cross-talk is the main Dot1 regulatory pathway. Not only the factors directly involved in H2B (de)ubiquitination were identified as regulators of H3K79 methylation, also new H2Bub modulators were discovered. One of these was Gcn5, which is responsible for the HAT activity in SAGA and was found to negatively regulate H3 methylation and H2Bub, at least in part by maintaining Ubp8 protein abundance. No candidate H3K79 demethylase was identified and histone deposition was found to negatively regulate H3K79 methylation, supporting the hypothesis that in the absence of a demethylase histone dilution is the main mechanism to counteract Dot1 activity. Several proteins involved in DNA repair through homologous recombination were scored as negative regulators of Dot1 activity, particularly proteins

required for the recovery of stalled replication forks through recombination. Finally, there are several candidate regulators that we have not yet studied further. These factors may regulate Dot1 indirectly, through one of the mechanisms described above, or through a yet to be discovered mechanism. Especially for these uncharacterized candidates, indirect effects of gene knock-outs on neighboring genes should also be considered (*Ben-Shitrit et al., 2012*). Future studies will be required to clarify the mechanisms of some of these regulators. The resource on Dot1 regulation presented here can be used as a starting point for such studies.

The Epi-ID screen reported here was performed using the knock-out collection of non-essential genes and offers a valuable resource for future studies on the regulation of H3K79 methylation in yeast and other organisms. However, Epi-ID is not limited to the use of knock-out collections but is highly flexible with regard to the library being screened. Combining barcode libraries with libraries of temperature sensitive alleles, histone mutants, or hypomorphic (DamP) alleles (*Ben-Aroya et al., 2008*; *Dai et al., 2008*; *Yan et al., 2008*) may lead to new insights into the contribution of essential genes in regulating chromatin structure and function. Dedicated, smaller custom libraries can also be made, as illustrated by the experiment testing the rescue potential of Gcn5-containing plasmids (*Figure 5B*). In this case one indexed Epi-ID experiment was an efficient and robust alternative to performing multiple replicate ChIPs and qPCR analyses.

The technology can easily be applied to chromatin features other than H3K79 methylation. As an example, we showed a small Epi-ID experiment for H2B ubiquitination (*Figure 5C*). Furthermore, a previous version of the technique has been used to identify histone turnover factors, albeit on a smaller scale (*Verzijlbergen et al., 2011*), and Epi-ID for H2A.Z levels has also been performed successfully (Korthout, van Leeuwen et al., submitted). The only requirement for an Epi-ID experiment is that a ChIP against the chromatin feature of interest can pull down one or both of the barcoded regions at the HO locus. We envision that it will be feasible to generate a library of clones with barcodes at a locus of interest in the near future, which would eliminate this limitation. Such strategies will allow for determining the regulome of many other chromatin features and investigating the cross-talk between the different networks.

Yeast offers the unique possibility to use genetic crosses and SGA technology to combine gene inactivations with chromatinized DNA barcodes. However, the concept of Epi-ID, i.e. barcode-ChIP-seq in mutant backgrounds, is not restricted to yeast and is in principle transferable to other organisms. The basic requirement is that mutants can be uniquely identified by a chromatinized DNA sequence, which is the case in shRNA, CRISPR and CRISPRi libraries when the genetic elements to inactivate genes are integrated in the genome (e.g. *Evers et al., 2016*). With the recent developments in genome targeting and editing, tools are becoming available to integrate barcoded libraries at a common locus to avoid position effects. Taken together, Epi-ID is a highly adaptable technique that enables the identification of new chromatin regulators.

## Materials and methods

### Yeast strains and plasmids

Yeast strains and plasmids used in this study are listed in *Supplementary file 1* and *Supplementary file 2*, respectively. Yeast media were described previously (*Tong and Boone, 2006*; *Van Leeuwen and Gottschling, 2002*). Library manipulations were done using the RoToR from Singer Instruments (Watchet, UK) and the synthetic genetic array (SGA) technology (*Tong and Boone, 2006*). The collection of barcoded knockouts was made by crossing a set of 1140 Barcoder strains (*Yan et al., 2008*) to each of five plates making up the *MAT*α NatMX knock-out collection (*Tong and Boone, 2006*). After mating, diploids were selected with G418 and CloNat double selection on rich media. After sporulation, the proper *MAT*a mutant strains were selected in several pinning steps: first on haploid *MAT*a selection (YC-His+Can+SAEC), then twice on *MAT*a double resistance selection (YC-His+Can+SAEC+MSG+G418+CloNat). The barcodes and deletions were checked for a few mutants and were found to be correct. Three extra Barcoder strains were generated by amplifying the barcoded KanMX locus from strains from the *MAT*a haploid gene knockout library (Open Biosystems, Huntsville, AL) and integrating it at the HO locus in strain BY4741. The *dot1Δ* and *bre1Δ* strains were taken from the NatMX library and a Dot1 over-expression strain was generated by integrating the *TDH3* promoter amplified from pYM-N15 (*Janke et al., 2004*) in front

of the *DOT1* gene in strain Y7092. These strains were crossed to the extra Barcoder strains to generate barcoded controls that have the same background as the library (NKI4557-4559). NKI4560 was generated by selecting a G418-resistant, CloNat-sensitive haploid after the library cross.

A custom set of barcoded deletion strains was made by re-arraying *MATα* NatMX deletion strains using the Stinger extension to the RoToR system (Singer Instruments), crossing these to Barcoders and selecting double-resistant *MAT*a haploids. Uniquely barcoded CloNat-sensitive *MAT*a haploids were used as wild-type strains. Strains were transformed with single-copy CEN-LEU2 plasmids using an adaptation of the microtiter plate LiAc protocol (*Gietz, 2014*). The Gcn5-containing plasmids were generated by replacing the TRP1 marker in pRS414-GCN5 and pRS414-gcn5-F221A (*van Oevelen et al., 2006*) by the LEU2 marker using homologous recombination in yeast.

NKI4657 was generated by integrating a barcoded KanMX at the HO locus of BY4733. In this background, several deletions were made for validation experiments. The endogenous 2μ plasmid was cured from some strains by transforming the pBIS-GALkFLP(URA3) plasmid into the cells (*Tsalik and Gartenberg, 1998*), inducing mutant FLP recombinase on galactose-containing medium and then selecting for the loss of the plasmid on media containing 5-fluoroorotic acid (FOA). Loss of the 2μ plasmid was confirmed by PCR. To test the role of the SUMO ligase activity of the essential Mms21, we truncated this essential gene at amino acid 183 to eliminate the catalytic domain, yet still support viability (*Zhao and Blobel, 2005*). This truncation was made by integrating a stop codon, *CYC1* terminator and NatMX cassette in the place of the C-terminal *MMS21* sequence.

Strains expressing FLAG-tagged (mutant) H2B, NKI4609 and NKI4610, were made by transforming pRG422 and pRG423, respectively, into NKI4602 and selecting cells that had lost the original plasmid.

To determine effects on the expression levels of H2Bub regulators, strains with TAP-tagged alleles were derived from the TAP-Fusion ORF collection (Dharmacon; *Ghaemmaghami et al., 2003*). *GCN5* and *NAT1* were replaced by NatMX in these strains. Because we could not retrieve a correct *RAD6*-TAP clone from the library, *RAD6* was TAP-tagged in BY4741 and derivates thereof, with the TAP-KanMX cassette amplified from pFvL29. Finally, because Bre1-TAP loses its interaction with Rad6 (*Wood et al., 2003*), we used KY2513, in which Bre1 carries a FLAG tag on the N terminus and deleted *GCN5* and *NAT1* in this strain.

## Epi-ID

Briefly, the Epi-ID screens consisted of five steps: (1) preparing chromatin from pools of cells, (2) Chromatin Immunoprecipitation experiments, (3) PCR reactions on purified DNA, (4) Sequencing, (5) Data analysis. The five plates of the collection were treated separately until after the PCR. The five plates were grown on rich media for one night and cells were scraped off and pooled in liquid media in the morning. The cultures were grown for 4–5 hr until in log phase and cells were cross-linked with formaldehyde, washed and harvested. Chromatin was prepared essentially as in *Vlaming et al. (2014)*, in the presence of SDS for H3C and methyl ChIPs, but sonication of chromatin from ~5E9 cells in 1.5 mL was done in 15 mL tubes. The ChIP was also performed as in *Vlaming et al. (2014)*, using polyclonal antibodies against H3K79me1, H3K79me3 and the H3 C terminus (*Frederiks et al., 2008*). In the small-scale Epi-ID experiments, also antibodies against H3K4me3 (RRID:AB_306649, lot GR273043-1) and H3K36me3 (RRID:AB_306966, lot GF260274-1) were used, as well as an H2BK123ub antibody that will be described elsewhere (manuscript in preparation). The UpTag and DownTag were amplified separately from the purified DNA (*Figure 1—figure supplement 1A*). The scale of each Epi-ID experiment was chosen such that on average an estimated number of 250 copies of each barcode was present in each PCR reaction to minimize jackpot effects (*Figure 1—figure supplement 1D*). The forward primer (AATGATACGGCGACCACCGAGATCTCGCTCTTCCGATCTAGATGTCCACGAGGTCTCT/AATGATACGGCGACCACCGAGATCTACACTCTTCCGATCTACGGTGTCGGTCTCGTAG for UpTag/DownTag) introduced the Illumina P5 sequence and extra nucleotides for annealing of the 5' end of the custom sequencing primers. With the reverse primer (CAAGCAGAAGACGGCATACGANNNNNNNGTCGACCTGCAGCGTACG/CAAGCAGAAGACGGCATACGANNNNNNAACGAGCTCGAATTCATCGA for UpTag/DownTag) the Illumina P7 sequence was introduced, as well as a 6-base-pair index. The uniquely barcoded amplicons were then mixed equimolarly and purified from an agarose gel with a size selection of 100–150 bp. The purified DNA was sequenced (single read, >50 bp) on a HiSeq2500 platform (Illumina, San Diego, CA) with High Output Run Mode, using a mix of custom sequencing primers for the UpTag and DownTag (CGCTC

TTCCGATCTAGATGTCCACGAGGTCTCT/    ACACTCTTCCGATCTACGGTGTCGGTCTCGTAG).
Since the above-mentioned primer sequences were not compatible with paired-end flow cells, the
oligonucleotide sequences were slightly altered (see *Supplementary file 3*) to sequence small-scale
Epi-ID experiments on a MiSeq (Illumina).

A Perl script, eXtracting Counting And LInking to Barcode References (xcalibr), was written to
transform raw sequencing reads to tables with counts for each index-barcode combination. In short,
the U2/D2 sequences were used to assign a read to UpTag or DownTag and the index sequence
behind the U2/D2 sequence and the barcode sequence in the beginning of the read were identified.
The xcalibr source code is available at https://github.com/NKI-GCF/xcalibr. In the counts table,
counts below ten were removed. After that, any barcode below 10% of the median in an input sam-
ple of a plate was considered not present on that plate and counts for all indices belonging to this
plate were removed. The tables were median-normalized for each index and converted to a table
with ORFs and experiments based on the barcode-index combinations. In this table IP columns could
be divided over each other or over input. H3K79me data of strains with H3/input of <0.5 was dis-
carded. Analysis for the small-scale experiments was different in the normalization step, where the
average wild-type count was set to 1.

## Growth rate determination and growth correction

Cells were grown up for an Epi-ID experiment, but cells were collected for gDNA isolation at two
time points prior to harvesting the cross-linked cells. Input material from chromatin made for the
Epi-ID experiment (not described in this study) was used for the third time point. gDNA was isolated
as described by *Hoffman and Winston (1987)*. The relative abundance of the barcodes at each time
was used to estimate growth rate for all the deletion strains, assuming a wild-type median growth
rate of 0.42 h$^{-1}$ (*Di Talia et al., 2007*) and using the formula N(t)=N0*e$^{\mu*t}$, where N(t) is the number
of cells at time t, N0 is the number of cells at t0 and μ is the growth rate. Growth rates that gave a
goodness of fit of <0.95 were discarded. Growth rates were calculated using data of the UpTag and
DownTag separately, and these values were averaged. A few cases were removed, where different
rates were calculated for the UpTag and DownTag (c.o.v. >0.2). Finally, the growth rates calculated
for two independent experiments were averaged. To create a growth-corrected H3K79 methylation
score, the log2-transformed me3/me1 value expected based on the fitness of the strain was sub-
tracted from the original log2-transformed me3/me1 value. The expected value was calculated using
the fits shown in *Figure 2A*.

## Immunoblotting

Quantitative immunoblotting was performed as described previously (*Vlaming et al., 2014*). Protein
extracts were made using NaOH or SUMEB lysis buffer. Chromatin samples were prepared as
described before (*Vlaming et al., 2014*), but without SDS, and incubated with Laemmli buffer at
95°C for 1 hr. The antibodies used in the Epi-ID experiment were also used on blots, as well as anti-
bodies against H3K79me2 (RRID:AB_1587126), Dot1 (*van Leeuwen et al., 2002*), H2B (39238,
Active Motif, Carlsbad, CA), Pgk1 (RRID:AB_221541), FLAG (RRID:AB_259529), TAP (RRID:AB_
10709700) and Sir2 (RRID:AB_656455).

## Mass spectrometry

Analysis of H3K79 methylation levels in the *gcn5Δ* strain was performed as described before
(*De Vos et al., 2011*), using multiple reaction monitoring (nanoLC-MRM) using a 4000 Q TRAP mass
spectrometer (AB SCIEX, Framingham, MA).

To measure H3K79 methylation levels in the DNA repair mutants, samples were prepared in the
same way as described before (*De Vos et al., 2011*). Samples were dissolved in 10% formic acid
prior to reverse phase nano-flow liquid chromatography on an EASY nLC 1000 system (Thermo
Scientific, San Jose, CA) coupled to a Thermo Orbitrap Fusion hybrid mass spectrometer (Thermo
Scientific. San Jose, CA). Briefly, peptides were separated on a ReproSil-Pur 120 C18-AQ 2.4 μm
(Dr. Maisch GmbH, Ammerbuch, Germany) 75 μm × 500 mm analytical column (packed in house) in
a 30 min. linear gradient from 10% to 40% solvent B (0.1% formic acid in 8:2 (v/v) acetonitrile:water)
followed by a 15-min wash at 100% solvent B at ~250 nl/min. Nanospray was achieved using the
Proxeon nanoflex source and fused silica gold coated emitters (pulled and coated in-house) at 1.55

kV. The mass spectrometer was configured in targeted mode to select precursor ions by quadrupole isolation at 1.6 Th, followed by HCD fragmentation with a normalized collision energy of 25 and Orbitrap MS2 fragment detection. The instrument was run in top speed mode with 3 s cycles.

PRM parameters were optimized using a set of four purified synthetic peptides with the sequence EIAQDFK*TDLR (K*: K-me0, -me1, -me2 and -me3). Briefly, doubly and triply charged precursors were fragmented in the Orbitrap Fusion and their four most abundant fragment ions were selected and validated using Skyline software (RRID:SCR_014080; *MacLean et al., 2010*). Label-free quantification was achieved by comparing the area of each methylation state to the sum of the areas of all methylation states. As the four different peptides have slightly different physicochemical properties and thus might have slightly different ionization efficiencies in the mass spectrometer, final peptide intensities were corrected using a relative response factor to obtain more accurate results. The relative response factor was obtained by measuring the peak areas of the four differently methylated peptide standards from an equimolar mixture and dividing peptide areas by the area of the unmethylated peptide.

## ChIP-qPCR

ChIP-qPCR experiments were performed as in *Vlaming et al. (2014)*, in the presence of SDS for IPs against H3 or H3K79 methylation. A *DC* protein assay (Bio-Rad, Hercules, CA) was used to quantify protein content of chromatin samples, with the purpose of equalizing the amount of chromatin going into each ChIP. Antibodies used were the same as for Epi-ID or immunoblot experiments, as well as an antibody against H3K9ac (RRID:AB_2118292, lot GR243602-1). Oligos used for qPCR can be found in *Supplementary file 3*. Each sample was measured in two technical duplicates in the qPCR and the average value of these two was taken as one value when combining biological replicates.

## SAM/SAH measurements

40–80M logarithmically-growing cells were pelleted and washed with cold TBS. Pellets were resuspended in 500 µL 5% perchloric acid and spun down after 30 min on ice, after which the supernatant was collected. SAM and SAH measurements were performed as described before (*Struys et al., 2000*). Briefly, SAM and SAH were extracted from yeast lysates using solid phase extraction (Oasis HLB, Waters, Milford, MA). Subsequently, the concentrations were determined using positive electrospray liquid chromatography tandem mass spectrometry (API5000, Applied Biosystems, Foster City, CA). The intra- and inter-assay CVs for SAM were 6.8% and 4.2%, respectively. The intra- and inter-assay CVs for SAH were 6.9% and 5.5%, respectively.

## Statistics

The online tool Cutoff Linked to Interaction Knowledge (CLIK) from the Rothstein laboratory was used to determine outlier groups and calculate enriched protein complexes (http://www.rothstein-lab.com/tools/clik; RRID:SCR_014690; *Dittmar et al., 2013*). The CLIK analysis relied on BioGRID version 3.4.130 (RRID:SCR_007393) for interaction data, and the curated list of protein complexes from *Baryshnikova et al. (2010)*. PANTHER (RRID:SCR_004869; *Mi et al., 2016*) was used to find enriched GO processes in a CLIK group.

Comparisons between wild-type and mutant strains for different methylation states were done using two-way ANOVA and corrected for multiple comparisons using the Šídák method, unless otherwise indicated. Sample sizes are reported in the figure legends and are always biological replicates, meaning that the cells were grown independently. All performed statistical tests can be found in *Supplementary file 4*. The significance thresholds used were $p < 0.05$ (*), $p < 0.01$ (**) and $p < 0.001$ (***). The asterisks indicate significant differences compared to wild type.

R (RRID:SCR_001905; *R Core Team, 2016*) and GraphPad Prism 6 (RRID:SCR_002798) were used for data analysis and plotting. Error bars represent standard deviation.

## Acknowledgements

We thank Brenda J Andrews, Charles Boone, Michael Costanzo and Corey Nislow for yeast collections, Karen M Arndt for the FLAG-Bre1 strain, H Th Marc Timmers for Gcn5 plasmids, Marc R Gartenberg for FLP plasmids, Richard G Gardner for FLAG-H2B plasmids and helpful discussions, and

Farid El Oualid (UbiQ), Alfred Nijkerk (UbiQ), Huib Ovaa and Reggy Ekkebus for help with the generation of the H2Bub antibody. We thank Virginia Rodriguez Martinez for help with initial Slx5/8 characterization, Gideon Oudgenoeg and Onno B Bleijerveld for help with mass spectrometry, Heinz Jacobs and Ron M Kerkhoven for discussions, and Bas van Steensel, Piet Borst and Eliza Mari Maliepaard for critical reading of the manuscript.

# Additional information

## Competing interests

FvL: The Netherlands Cancer Institute and FvL are entitled to royalties that may result from licensing the yeast H2BK123ub-specific monoclonal antibody according to IP policies of the Netherlands Cancer Institute. The other authors declare that no competing interests exist.

## Funding

| Funder | Grant reference number | Author |
| --- | --- | --- |
| Nederlandse Organisatie voor Wetenschappelijk Onderzoek | NWO-VICI-016.130.627 | Fred van Leeuwen |
| KWF Kankerbestrijding | KWF NKI2014-7232 | Fred van Leeuwen |
| KWF Kankerbestrijding | KWF NKI2009-4511 | Fred van Leeuwen |
| Nederlandse Organisatie voor Wetenschappelijk Onderzoek | NWO-VIDI 723.012.102 | AF Maarten Altelaar |
| National Roadmap Large-scale Research Facilities of The Netherlands | 184.032.201 | Liesbeth Hoekman AF Maarten Altelaar |
| Nederlandse Organisatie voor Wetenschappelijk Onderzoek | NCI-KIEM-731.013.102 | Fred van Leeuwen |
| Nederlandse Organisatie voor Wetenschappelijk Onderzoek | NCI-LIFT-731.015.405 | Fred van Leeuwen |

The funders had no role in study design, data collection and interpretation, or the decision to submit the work for publication.

## Author contributions

HV, FvL, Conception and design, Acquisition of data, Analysis and interpretation of data, Drafting or revising the article; TMM, TvW, Acquisition of data, Analysis and interpretation of data; DWP-L, SH, Acquisition of data; DES, Performed and analyzed metabolite measurements; AV, Developed the xcalibr script to process sequencing reads; LH, Performed and analyzed mass spectrometry measurements; TK, Analysis and interpretation of data; AFMA, Resonsible for mass spectrometry measurements and analysis

## Author ORCIDs

Hanneke Vlaming, http://orcid.org/0000-0003-1743-6428
Fred van Leeuwen, http://orcid.org/0000-0002-7267-7251

# Additional files

## Supplementary files

- Supplementary file 1. Yeast strains used in this study.

- Supplementary file 2. Plasmids used in this study.

- Supplementary file 3. Primers used in the study.

- Supplementary file 4. Statistical tests performed in the study.

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
