## [Decision Letter]

Thank you for submitting your article "Direct screening for chromatin status on DNA barcodes in yeast delineates the regulome of H3K79 methylation by Dot1" for consideration by *eLife*. Your article has been favorably evaluated by Jessica Tyler (Senior Editor) and three reviewers, one of whom is a member of our Board of Reviewing Editors. The reviewers have opted to remain anonymous.

The reviewers have discussed the reviews with one another and the Reviewing Editor has drafted this decision to help you prepare a revised submission.

Summary:

The authors describe a new strategy for carrying out an unbiased screen for genes that regulate chromatin modifications. The Epi-ID method allows pools of individual mutants, each carrying a specific integrated barcode, to be probed by ChIP. Deep sequencing of the barcodes in the precipitated samples allows quantitative assessment of the effects of a given mutation on the chromatin parameter, in this case H3K79me1 and H3K79me3. In addition to the genes previously known to regulate H3K79 methylation, either directly or indirectly, the authors identify novel genes including the Gcn5 acetyltransferase, N-terminal acetyltransferases, genes that regulate replication fork recovery, and enzymes involved in S-adenosyl methionine (SAM) and SAH biosynthesis. The manuscript also makes a strong case that there is no H3K79 demethylase, with the caveat that essential strains were not tested. Overall, the reviewers found the Epi-ID method to have strong potential for identifying novel genes impacting methylation that could be readily extended to other chromatin modifications for which the appropriate antibodies are available. However, a more thorough exploration of some of the hits would make a stronger case of the utility of the approach.

Essential revisions:

1) The interpretation of Figure 4 is speculative, and the results on the leukemia cell lines do not clarify the conclusions. The adenosine kinase inhibitors could be influencing cell viability through a mechanism quite unrelated to Dot1L or H3K79me. What do the H3K79me levels look like in the treated cells? In their current form, these tissue culture experiments weaken the paper and should be omitted unless further pursued.

2) The finding that deletion of ADO1 impacts histone methylation is presented as a discovery, whereas it seems to fall in the category of a confirmation of the validity of the screen rather than new information. Since ADO1 was already reported to regulate SAM levels (Kanai 2013), it is not surprising that altering levels of the methyl donor would impact histone methylation. It was initially puzzling that deletion of ADO1 had opposing effects on Dot1 versus Set1 and Set2. However, the explanation that the authors suggest, namely that these enzymes respond differently to restriction of SAM and SAH is, in fact, consistent with the cited paper (Sadhu et al., 2013), which reported differential effects on H3K79 versus H3 K4 methylation in response to nutritional limitation. The lengthy discussion of this gene does not seem warranted in light of what was previously known. Instead, this could be cast differently, namely as support for the Epi-ID method.

3) One of the more intriguing findings is the observed impact of acetyltransferase mutations on H3K79 methylation via an effect on H2B ubiquitination, which is required for methylation by Dot1. A deletion or catalytic mutant of the Gcn5 acetyltransferase increased methylation via an impact on H2B ubiquitination, whereas mutations in the N-terminal acetyltransferase proteins, Nat1 and Ard1, caused a decrease in H2B ubiquitination. However, these results are not pursued adequately, leaving open the question of whether these genes directly regulate H2B ubiquitination or have indirect effects. At minimum, the authors should show whether any of the acetyltransferase mutants affect levels of proteins known to be involved in H2B ubiquitination, Bre1, Rad6 and Lge1, or deubiquitination, Ubp8 and Ubp10. Potential effects on protein levels should be explored for any other mutants discussed, as was done for Slx5/8.

4) Do the HAT mutations also affect H3K4 methylation?

5) The western blot in Figure 4 for Ado1 effects is not particularly convincing. H3K4me3 and H3K36me3 levels also seem to be decreased.

6) What do H2Bub levels look like in the ado1 mutant?

7) Conclusions drawn from Figure 1 (even in regards to controls) should be qualified in regards to effects of deletions on growth rates which affect K79me3 (as the authors address in Figure 2).

8) How do the authors ensure that the time points in Figure 2 represent log phase growth? While the growth rates for two independent experiments were determined and then averaged, can the same conclusions about K79me3 and growth rate be derived independently from each experiment? Assuming this is the case, this should be stated (although it may be buried in Figure 2 legend). It would also be helpful to make clear in the main text that the strains were separately grown, as opposed to grown as a pool.

[Editors' note: further revisions were requested prior to acceptance, as described below.]

Thank you for resubmitting your work entitled "Direct screening for chromatin status on DNA barcodes in yeast delineates the regulome of H3K79 methylation by Dot1" for further consideration at *eLife*. Your revised article has been favorably evaluated by Jessica Tyler (Senior editor), a Reviewing editor, and 1 reviewer.

The manuscript has been improved but there are some remaining issues that need to be addressed before acceptance, as outlined below:

The revised manuscript appropriately addresses almost all previous concerns. The addition of controls in which protein levels were measured in the various mutant strains improved the manuscript and helped clarify interpretations. The reduced emphasis on Ado1, along with a discussion of these results in the context of published studies, also improved the paper. Finally, the removal of the cell line experiments, which were incomplete, solidified the manuscript.

One remaining issue in the revised manuscript is that Gcn5, and Gcn5 catalytic activity, are described as regulators of Ubp8 activity. However, Ubp8 protein levels are reduced in the gcn5 deletion strain, indicating that the defect in H2B deubiquitylation is a consequence of the loss of the SAGA subunit, Ubp8, rather than regulation of its catalytic activity. The authors show that a mutant of Gcn5, F221A, has a phenotype comparable to the deletion; however, F221 forms hydrophobic interactions with multiple buried residues, which raises the possibility that the strong effect of the F221A mutation results from destabilization of Gcn5, which would be predicted to lead to loss of Ubp8. To demonstrate that Gcn5 indeed regulates Ubp8 activity, as opposed to stabilizing its incorporation into SAGA, the authors should provide evidence that the levels of both Gcn5 and Ubp8 are unchanged when Gcn5 bears the F221A substitution. Without this result, the authors should soften their conclusions here. Should the authors wish to generate a bona fide Gcn5 catalytic mutant, substitution of the conserved catalytic E122 would be a better choice.

---

## [Author Response]

Essential revisions:

1) The interpretation of Figure 4 is speculative, and the results on the leukemia cell lines do not clarify the conclusions. The adenosine kinase inhibitors could be influencing cell viability through a mechanism quite unrelated to Dot1L or H3K79me. What do the H3K79me levels look like in the treated cells? In their current form, these tissue culture experiments weaken the paper and should be omitted unless further pursued.

The tissue culture experiments were included to indicate the possible utility or relevance of the Epi-ID results for human biology. However, we agree with the comments that those results are still preliminary; taking the cell line part to the same level as the rest of the manuscript would require substantially more work and therefore falls beyond the main scope of this paper. We therefore followed your suggestion to omit the cell culture experiments from the manuscript and instead focused on additional experiments to strengthen the other findings (see below).

2) The finding that deletion of ADO1 impacts histone methylation is presented as a discovery, whereas it seems to fall in the category of a confirmation of the validity of the screen rather than new information. Since ADO1 was already reported to regulate SAM levels (Kanai 2013), it is not surprising that altering levels of the methyl donor would impact histone methylation. It was initially puzzling that deletion of ADO1 had opposing effects on Dot1 versus Set1 and Set2. However, the explanation that the authors suggest, namely that these enzymes respond differently to restriction of SAM and SAH is, in fact, consistent with the cited paper (Sadhu et al., 2013), which reported differential effects on H3K79 versus H3 K4 methylation in response to nutritional limitation. The lengthy discussion of this gene does not seem warranted in light of what was previously known. Instead, this could be cast differently, namely as support for the Epi-ID method.

We thank the reviewers for the suggestion to adjust the way in which the phenotype of the ADO1 knock-out is presented. Taking into account your comments and the results we obtained while addressing comments #5 and #6, we shortened the discussion of this gene and moved the Results section more towards the beginning of the manuscript where the regulators are discussed that confirm the power of the screening method (subsection “Adenosine kinase promotes histone methylation”). We have made a new Figure 3, which describes both positive regulators, the NatA complex and Ado1.

We also clarified why we believe the observed changes in the ADO1 knock-out are unexpected and do not just confirm what has previously been shown. Briefly, it has been reported that both SAM and SAH levels are up in ado1d cells, but the extent of the SAH increase was not reported in previous studies. Therefore, it was not known how the SAM/SAH ratio was affected and the prediction would be that it would decrease, given where it acts on the methionine cycle. The SAM/SAH ratio is generally considered to represent the methylation potential in the cell, so also based on the reduced histone H3 methylation that we observed a reduction in the SAM/SAH ratio was expected. However, we found that the increase in SAM is actually larger than the increase and SAH, leading to a higher SAM/SAH ratio. Therefore, our results suggest that the SAM/SAH ratio cannot be taken as the sole determinant of methylation potential and that absolute levels of SAM and SAH should also be taken into consideration. We now discuss this more clearly in the last paragraph of the aforementioned subsection.

The study by Sadhu et al. (2013) nicely shows how Dot1 and Set1 seem to respond differently to perturbation of folate synthesis and supply in yeast; we now cite this paper in the Results section to emphasize the role of nutrient levels on histone methylation. However, the SAM levels were not altered in a folate synthesis mutant (fol3Δ), whereas deletion of ADO1 causes large changes in SAM and SAH. Considering this difference, we prefer to not directly compare the histone methylation phenotypes in ado1Δ and fol3Δ mutants and focus the discussion of the results on the SAM and SAH changes.

3) One of the more intriguing findings is the observed impact of acetyltransferase mutations on H3K79 methylation via an effect on H2B ubiquitination, which is required for methylation by Dot1. A deletion or catalytic mutant of the Gcn5 acetyltransferase increased methylation via an impact on H2B ubiquitination, whereas mutations in the N-terminal acetyltransferase proteins, Nat1 and Ard1, caused a decrease in H2B ubiquitination. However, these results are not pursued adequately, leaving open the question of whether these genes directly regulate H2B ubiquitination or have indirect effects. At minimum, the authors should show whether any of the acetyltransferase mutants affect levels of proteins known to be involved in H2B ubiquitination, Bre1, Rad6 and Lge1, or deubiquitination, Ubp8 and Ubp10. Potential effects on protein levels should be explored for any other mutants discussed, as was done for Slx5/8.

Following the suggestions, we performed a series of experiments to obtain more insight into the regulatory roles of the acetyltransferase mutants:

We checked the protein levels of Dot1 for both nat1d and gcn5d strains (Figure 3—figure supplement 1 and Figure 5 and Figure 5—figure supplement 2, respectively) and found that neither the NatA complex nor Gcn5 affect the expression of Dot1 itself.

We checked the protein levels of positive H2Bub regulators Bre1, Rad6, Lge1, Rtf1, and negative H2Bub regulators Ubp8 and Ubp10, for both nat1d and gcn5d strains (Figure 3—figure supplement 1 and Figure 5, respectively). No significant decrease in H2B ubiquitination factors or increase in H2Bub deubiquitinating factors was observed in nat1d strains (Figure 3—figure supplement 1), as discussed in the last paragraph of the subsection “The NatA complex regulates H3K79 methylation and H2B ubiquitination”.

In gcn5d strains, of all the H2Bub regulators, only Ubp8 showed a significant change in protein expression that could explain the observed change in H2Bub (Figure 5). We observed this decrease in whole-cell extracts and confirmed it in (independent) chromatin fractions (Figure 5—figure supplement 2). These findings suggest that Gcn5 is required for Ubp8 stability, potentially by regulating the association with its partners in the SAGA DUB module, as discussed in the last paragraph of the subsection “The SAGA HAT module negatively regulates histone methylation and H2B ubiquitination”.

To strengthen the result that Gcn5 regulates H2Bub, we also performed a ChIP-qPCR experiment on several loci, which is shown in Figure 5 and Figure 5—figure supplement 3. This validated the H2Bub increase in gcn5d strains. Moreover, while analyzing protein expression in WT and gcn5d strains (described above), we again repeatedly observed the global increase in H2Bub in gcn5d strains. This effect can now also be observed in Figure 5 and Figure 5—figure supplement 2.

Dot1 immunoblots have now also been included for the ado1 mutants (Figure 3—figure supplement 2).

4) Do the HAT mutations also affect H3K4 methylation?

This is a very interesting question and we decided to study the effects on both H3K4me3 and H3K36me3 for Gcn5 and the NatA complex.

For the NatA complex we addressed this question by immunoblot analysis, since the effects on H2Bub and H3K79me can easily be measured on a global level. As expected, given the H2Bub effect, H3K4me3 was lost in bre1d cells and decreased in nat1d or ard1d cells (Figure 3). We also noticed that H3K36me3 was decreased in the absence of Bre1 (Figure 3), which to our knowledge has not been described before in literature. We validated the effect of H2Bub on H3K36me3 (Figure 3—figure supplement 1) and also found a partial effect of the NatA complex on H3K36me3, consistent with its partial H2Bub loss (Figure 3).

To further investigate the role of Gcn5, which has smaller global H3K79me effects, we decided to perform a new small-scale Epi-ID experiment using strains that contained empty plasmid or a wild-type/mutant Gcn5-expressing plasmid. In this experiment we used antibodies against H3K4me3 and H3K36me3, as well as antibodies against H2Bub, H3K79me1 and H3K79me3. We observed the same H3K79me methylation effects as before on the barcoded locus (the previous results have now been moved to the figure supplement and the new result can be found in Figure 5) and observed an increase in H2B ubiquitination on this locus in a gcn5d strain. This regulation of H2Bub levels was dependent on the catalytic activity of Gcn5. Finally, both H3K4me3 and H3K36me3 were significantly increased in the absence of Gcn5 or its catalytic activity.

These additional experiments show that, NatA and Gcn5 both affect H2B ubiquitination and the trimethylated states of H3K4, H3K36 and H3K79, which are all promoted by H2Bub.

5) The western blot in Figure 4 for Ado1 effects is not particularly convincing. H3K4me3 and H3K36me3 levels also seem to be decreased.

We previously observed that the decrease in H3K79me3 was somewhat stronger and more reproducible than the decreases in H3K4me3 and H3K36me3, which led us to conclude that Dot1 was more affected by Ado1 than Set1 and Set2. This was based on the quantification of three replicates, of which only one was shown. To address the variation between replicates (see also response to comment #6), we repeated these experiments and performed additional immunoblots. The old blots are now shown in Figure 3—figure supplement 2, the new results are shown in Figure 3 and Figure 3—figure supplement 2. Considering the newly obtained data, we conclude that the decreases in H3K79me3, H3K4me3 and H3K36me3 are comparable. The text has been adjusted accordingly (subsection “Adenosine kinase promotes histone methylation”, first paragraph).

6) What do H2Bub levels look like in the ado1 mutant?

To address this question we first performed an H2B blot on the samples that were used for the methylation blots originally shown in the main figure (now Figure 3—figure supplement 2). We found that the lower H3K79me levels in the ado1d strain could not be explained by lower H2Bub levels. Instead, the blots suggested that H2B ubiquitination was increased in the ado1d strain, potentially even masking stronger methylation defects. To confirm this unexpected result, we generated additional biological replicates of three independently generated wild-type and *ADO1* mutant strains. Collectively, the results show that all three methyltransferases (Set1, Set2 and Dot1) are affected by loss of Ado1, and that there is small increase in H2B ubiquitination, which could partially mask stronger defects in methyltransferase activity caused by altered SAM/SAH levels. We discuss these results in the first paragraph of the subsection “Adenosine kinase promotes histone methylation”.

7) Conclusions drawn from Figure 1 (even in regards to controls) should be qualified in regards to effects of deletions on growth rates which affect K79me3 (as the authors address in Figure 2).

In the Results section, we have now added the following sentence after explaining Figure 2: “It can also be appreciated from this plot that the known regulators identified in Figure 1 remain outliers after taking into account the growth defects that some of them have.”

8) How do the authors ensure that the time points in Figure 2 represent log phase growth? While the growth rates for two independent experiments were determined and then averaged, can the same conclusions about K79me3 and growth rate be derived independently from each experiment? Assuming this is the case, this should be stated (although it may be buried in Figure 2 legend). It would also be helpful to make clear in the main text that the strains were separately grown, as opposed to grown as a pool.

First, to clarify, growth rates were determined in pools of cells, making use of barcode sequencing of DNA isolated at each of the time points. Details can be found in the Methods section (subsection “Growth rate determination and growth correction”), but we have adjusted the text in the Results section to make this more clear (subsection “Deposition of new histones counteracts H3K79 methylation in the absence of a demethylase”, first paragraph).

The growth rates were determined in experiments separate from the H3K79me regulator Epi-ID screens. We have now phrased this more clearly in the Methods section (subsection “Growth rate determination and growth correction”). We have also included a plot (Figure 2—figure supplement 1) to show the strong correlation between the growth rates determined in the two replicate experiments, but cannot compare one of the growth rate replicates with one of the H3K79me replicates.

In the growth rate experiments, we also determined cell densities by OD measurement for each pool of cells at each time point. In Figure 2—figure supplement 1 we have now included a plot of these densities to show that the increases in cell numbers fit logarithmic cell growth throughout the course of the experiment. However, since the OD measurements are not very precise and there was one clear outlier, we used the more robust internal normalization (i.e. assuming a median growth rate of 0.42h^-1^) to calculate the growth rates.

[Editors' note: further revisions were requested prior to acceptance, as described below.]

The manuscript has been improved but there are some remaining issues that need to be addressed before acceptance, as outlined below:

The revised manuscript appropriately addresses almost all previous concerns. The addition of controls in which protein levels were measured in the various mutant strains improved the manuscript and helped clarify interpretations. The reduced emphasis on Ado1, along with a discussion of these results in the context of published studies, also improved the paper. Finally, the removal of the cell line experiments, which were incomplete, solidified the manuscript.

One remaining issue in the revised manuscript is that Gcn5, and Gcn5 catalytic activity, are described as regulators of Ubp8 activity. However, Ubp8 protein levels are reduced in the gcn5 deletion strain, indicating that the defect in H2B deubiquitylation is a consequence of the loss of the SAGA subunit, Ubp8, rather than regulation of its catalytic activity. The authors show that a mutant of Gcn5, F221A, has a phenotype comparable to the deletion; however, F221 forms hydrophobic interactions with multiple buried residues, which raises the possibility that the strong effect of the F221A mutation results from destabilization of Gcn5, which would be predicted to lead to loss of Ubp8. To demonstrate that Gcn5 indeed regulates Ubp8 activity, as opposed to stabilizing its incorporation into SAGA, the authors should provide evidence that the levels of both Gcn5 and Ubp8 are unchanged when Gcn5 bears the F221A substitution. Without this result, the authors should soften their conclusions here. Should the authors wish to generate a bona fide Gcn5 catalytic mutant, substitution of the conserved catalytic E122 would be a better choice.

We have made two adjustments to address this issue.

First, it was not our intention to suggest that Gcn5 (and its catalytic activity) are regulators of Ubp8 activity. We did not state this point in the manuscript. In fact, we show that the level of Ubp8 is reduced in the absence of wild-type Gcn5, suggesting that the effect on H2Bub is indeed more likely to be mediated by altered Ubp8 levels rather than altered Ubp8 activity. To our knowledge, a role for Gcn5 in modulating Ubp8 stability has not been reported previously. To make this clearer and avoid any confusion, we adjusted the text, where we, for example, now explicitly mention that ‘we examined the expression of H2Bub regulators and observed that Ubp8 showed a significant change in protein expression in the absence of Gcn5… A reduction in Ubp8 protein level was also observed in chromatin fractions… These findings suggest that Gcn5 is required for Ubp8 stability, potentially by regulating the association with its partners in the SAGA DUB module’.

Second, we chose the Gcn5-F221A mutant because it had previously been used by several other labs to determine the role of the catalytic activity of Gcn5; the mutation did not seem to affect the expression of Gcn5, the Gcn5-F221A protein still interacted with Ada2, and it acted in a dominant negative fashion, together indicating that the loss of activity caused by the F221A mutation was not due to Gcn5 destabilization (e.g. Kuo G&D 1998; Wang G&D 1998; Li MCB 2005; van Oevelen JBC 2006; Lanza PLoSOne 2012). However, the reviewer is absolutely right that we cannot exclude that the mutation has additional effects on Gcn5 protein folding or stability, especially given the location of the F221 residue. We thank the reviewer for providing this expert insight and have made a few adjustments in the text (subsection “The SAGA HAT module negatively regulates histone methylation and H2B ubiquitination”, first paragraph, and the Abstract, Introduction, fifth paragraph, and subsection “The SAGA HAT module negatively regulates histone methylation and H2B ubiquitination”, second and fifth paragraphs) to discuss this possibility and soften our conclusions about the role of the catalytic activity. The main section now ends with ‘This finding suggests that H3K79me regulation by Gcn5 depends on its catalytic activity. However, it cannot be excluded that the F221A mutation acts by other mechanisms such as altering nucleosome binding or the stability or protein folding of Gcn5.’